# Novel Salinomycin-Based Paramagnetic Complexes—First Evaluation of Their Potential Theranostic Properties

**DOI:** 10.3390/pharmaceutics14112319

**Published:** 2022-10-28

**Authors:** Irena Pashkunova-Martic, Rositsa Kukeva, Radostina Stoyanova, Ivayla Pantcheva, Peter Dorkov, Joachim Friske, Michaela Hejl, Michael Jakupec, Mariam Hohagen, Anton Legin, Werner Lubitz, Bernhard K. Keppler, Thomas H. Helbich, Juliana Ivanova

**Affiliations:** 1Department of Biomedical Imaging and Image-Guided Therapy, Division of Molecular and Structural Preclinical Imaging, Preclinical Imaging Laboratory, Medical University of Vienna & General Hospital of Vienna, Waehringer Guertel 18–20, 1090 Vienna, Austria; 2Institute of General and Inorganic Chemistry, Bulgarian Academy of Sciences, Akad. G. Bonchev Str., bl. 11, 1113 Sofia, Bulgaria; 3Faculty of Chemistry and Pharmacy, Sofia University “St. Kliment Ohridski”, J. Bourchier Blvd., 1, 1164 Sofia, Bulgaria; 4Chemistry Department, R&D, BIOVET Ltd., 39 Peter Rakov Str., 4550 Peshtera, Bulgaria; 5Institute of Inorganic Chemistry, University of Vienna, Waehringer Strasse 42, 1090 Vienna, Austria; 6Department of Inorganic Chemistry—Functional Materials, University of Vienna, Waehringer Strasse 42, 1090 Vienna, Austria; 7BIRD-C GmbH, Dr. Bohrgasse 2–8, 1030 Vienna, Austria; 8Faculty of Medicine, Sofia University “St. Kliment Ohridski”, Kozjak Str., 1, 1407 Sofia, Bulgaria

**Keywords:** theranostics, paramagnetic salinomycin complexes, bacterial ghosts, gadolinium, manganese, MRI

## Abstract

Combining therapeutic with diagnostic agents (theranostics) can revolutionize the course of malignant diseases. Chemotherapy, hyperthermia, or radiation are used together with diagnostic methods such as magnetic resonance imaging (MRI). In contrast to conventional contrast agents (CAs), which only enable non-specific visualization of tissues and organs, the theranostic probe offers targeted diagnostic imaging and therapy simultaneously. Methods: Novel salinomycin (Sal)-based theranostic probes comprising two different paramagnetic metal ions, gadolinium(III) (Gd(III)) or manganese(II) (Mn(II)), as signal emitting motifs for MRI were synthesized and characterized by elemental analysis, infrared spectral analysis (IR), electroparamagnetic resonance (EPR), thermogravimetry (TG) differential scanning calorimetry (DSC) and electrospray ionization mass spectrometry (ESI-MS). To overcome the water insolubility of the two Sal-complexes, they were loaded into empty bacterial ghosts (BGs) cells as transport devices. The potential of the free and BGs-loaded metal complexes as theranostics was evaluated by in vitro relaxivity measurements in a high-field MR scanner and in cell culture studies. Results: Both the free Sal-complexes (Gd(III) salinomycinate (Sal-Gd(III) and Mn(II) salinomycinate (Sal-Mn(II)) and loaded into BGs demonstrated enhanced cytotoxic efficacy against three human tumor cell lines (A549, SW480, CH1/PA-1) relative to the free salinomycinic acid (Sal-H) and its sodium complex (Sal-Na) applied as controls with IC_50_ in a submicromolar concentration range. Moreover, Sal-H, Sal-Gd(III), and Sal-Mn(II) were able to induce perturbations in the cell cycle of treated colorectal and breast human cancer cell lines (SW480 and MCF-7, respectively). The relaxivity (*r*_1_) values of both complexes as well as of the loaded BGs, were higher or comparable to the relaxivity values of the clinically applied contrast agents gadopentetate dimeglumine and gadoteridol. Conclusion: This research is the first assessment that demonstrates the potential of Gd(III) and Mn(II) complexes of Sal as theranostic agents for MRI. Due to the remarkable selectivity and mode of action of Sal as part of the compounds, they could revolutionize cancer therapy and allow for early diagnosis and monitoring of therapeutic follow-up.

## 1. Introduction

Cancer theranostics, the combination of a diagnostic with a therapeutic, pave the way for personalized medicine, allowing for simultaneous and precise diagnosis and treatment of malignant diseases. Current theranostic concepts aim at (i) facilitating the observation and monitoring of anticancer drugs, (ii) providing targeted and personalized cancer therapy, and (iii) realizing simultaneous diagnosis and treatment in (early-stage) cancer patients [1]. Magnetic Resonance Imaging (MRI) is a powerful non-invasive diagnostic technique, which takes advantage of a very high spatial and temporal resolution and can provide detailed molecular/cellular information when combined with a contrast agent with high relaxivity to overcome the lack of sensitivity inherent to MRI [2]. In contrast to conventional contrast agents (CAs), which enable only non-specific visualization of tissues and organs, the theranostic probe offers targeted diagnostic imaging and therapy simultaneously [3]. Presently, there is only one clinically approved theranostic agent for the treatment of progressive prostate cancer, recently approved by the FDA [4]. Therefore, the development of efficient, tumor-directed drugs with good magnetic susceptibility and a higher safety profile remains the main objective in modern oncology.

The design of new CAs plays an essential role in contrast-enhanced MRI today. Currently used CAs are predominantly low-molecular-weight gadolinium(III) (Gd(III))-based complexes, which can provide outstanding positive MR images with high resolution [5]. Unfortunately, some investigations have shown that Gd(III) could be involved in nephrogenic systemic fibrosis (NSF), which limits its clinical applications [6]. Newer studies, however, report the accumulation of Gd(III) in various tissues (bone, brain, and kidneys) of patients who were not diagnosed with renal impairment [7,8]. Recent toxicity concerns associated with the long-term use of low-molecular-weight acyclic Gd-based contrast agents (GBCA) have resulted in the restriction of their administration by the European Medicines Agency (EMA) and have triggered risk warnings from the U.S. Food and Drug Administration (FDA) [9,10]. Therefore, novel chelators, on the one hand, as well as alternative paramagnetic centers, on the other, as part of the MRI CAs, are needed urgently.

Due to its favorable properties, such as high spin number, long electronic relaxation time, and labile water exchange, manganese(II) (Mn(II)) represents an attractive alternative metal center for the design of novel MRI CAs [11]. The natural polyether antibiotic Salinomycin (Sal) has attracted the attention of numerous scientists all over the world as a highly selective cytotoxic agent and has been the subject of intense investigations during the past few years [12,13,14,15,16,17,18,19]. The distinctive pentacyclic spiroketal ring system that characterizes the molecule, and the ample presence of oxygen atoms from different functional groups (carboxylic, carbonyl, ether, hydroxyl), makes it a potential ligand able to bind paramagnetic metal cations such as Gd(III) or Mn(II). Therefore, it is expected to provide better stability and a safer profile. Sal has been reported to form complexes with metal(II) ions such as Zn^2+^, Cu^2+^, Co^2+^, and Ni^2+^, which exerted superior anticancer activity in vitro compared to the free ligand [20,21]. To the best of our knowledge, there are no data on the coordination of the antibiotic to trivalent metal ions.

Bacterial ghost cells (BGs) represent empty cell envelopes of Gram-negative bacteria devoid of cytoplasmic content and free of nucleic acids. They are produced by the controlled expression of the plasmid-encoded lysis gene E. Protein E, which leads to the fusion of the inner and outer membranes of the bacteria and the formation of a tunnel structure through which the cytoplasmic content is expelled as a result of the osmotic pressure difference between the cytoplasm and the exterior [22]. The BGs can be used as carriers or targeting vehicles for active agents, which are specific to various types of tissue. Moreover, active agents are efficiently transported to the desired destination since it is possible to prepare BGs that contain only the desired active substance and a high degree of loading, and thus, high efficiency of the active agent can be achieved. When BGs loaded with an active agent (in particular, a diagnostic, therapeutic, or theranostically active agent) are administered, they are internalized by cancer cells, followed by degradation of the BGs within the cancer cells. Thereupon, the active agent is released into the cytoplasm of the cancer cells, inducing cell death of the cancer cells [23,24].

Herein, we report, for the first time, the evaluation of Sal complexes with the paramagnetic ions Gd(III) (Sal-Gd(III)) and Mn(II) (Sal-Mn(II)) and their incorporation into BGs as effective theranostic agents for the early diagnosis and monitoring of cancer therapy with MRI.

## 2. Methods

### 2.1. Chemicals

The commercially available pharmaceutical-grade salinomycin sodium (C_42_H_69_O_11_Na; SalNa) was provided by Biovet Ltd. (Peshtera, Bulgaria), purity of >95%. Salinomycinic acid was prepared as previously reported [21]. Organic solvents (MeCN, MeOH, DMSO) and metal salts of analytical grade (GdCl_3_ and MnCl_2_.4H_2_O) were purchased from Fisher Scientific (Loughborough, UK). The tetrazolium salt 3-(4,5-dimethylthiazol-2-yl)-2,5-diphenyl tetrazolium bromide (MTT) was bought from Sigma-Aldrich (Vienna, Austria).

### 2.2. Synthesis of Paramagnetic Complexes of Sal

#### 2.2.1. Synthesis of Gd(III) Salinomycinate

The metal salt (GdCl_3_.6H_2_O, 0.1742 g, 0.47 mmol) was dissolved in 2 mL water. The solution was added to salinomycin sodium solution (0.2561 g, 0.33 mmol, dissolved in MeCN:MeOH = 1:5). The reaction mixture was stirred for 30 min at room temperature. After the slow evaporation of the solvents for seven days, a white precipitate was formed. The solid phase was washed with water, filtered off, and dried over P_4_O_10_. Yield: 221 mg, 82%. Anal. Calcd. for C_126_H_213_O_36_Gd (MW = 2458.25 g/mol): H, 8.65%; C, 61.43%; Gd, 6.39%. Found: H, 8.74%; C, 61.61%; Gd 6.28%. ESI-MS, m/z: 906.76 [(C_42_H_68_O_11_)Gd]^+^, calcd: 906.40 (100%), 908.40 (88.0%), 904.40 (82.4%), 905.40 (63%); 1657.56 [C_42_H_69_O_11_)_2_Gd]^+^, calcd: 1656.89 (100%); 1657.90 (90.9%); 1658.90 (88%); 1654.89 (82.4%); 1659.90 (80%); 1655.89 (74.9%); 1655.89 (63%).

#### 2.2.2. Synthesis of Mn(II) Salinomycinate

Salinomycin sodium (0.3850 g, 0.5 mmol) was dissolved in mixed solvents (20 mL MeOH + 2 mL MeCN). The solution of Mn(II) chloride (MnCl_2_.4H_2_O, 1 mmol, 198 mg in 4 mL water) was added to the ligand solution. The reaction mixture was stirred at room temperature for 30 min. The solution slowly evaporated, and the resulting light-brownish precipitate was washed with water, filtered off, and dried over P_4_O_10_ for three days. Yield: 329 mg, 83%. Anal. Calcd. for C_84_H_142_O_24_Mn (MW = 1590.94 g/mol): H, 8.93%; C, 63.36%; Mn, 3.45%. Found: H, 9.35%; C, 63.05%; Mn, 3.24%. ESI-MS, m/z: 804.76 [(C_42_H_69_O_11_)Mn]^+^, calcd.: 804.42 (100%); 805.43 (45.4%); 1578.55 [(C_42_H_69_O_11_)_2_MnNa]^+^, calcd: 1576.90 (100%); 1577.90 (90.9%); 1578.90 (40.8%).

The Gd(III) and Mn(II) complexes of salinomycin are insoluble in water and soluble in organic solvents, such as MeOH, MeCN, EtOH, CHCl_3_, DMSO, and hexane.

#### 2.2.3. Loading of Gd(III)and Mn(II) Salinomycinates into Empty Bacterial Ghosts Cells (BGs)

Non-pathogenic *Escherichia coli Nissle* 1917 (*EcN*1917) and *Escherichia coli NM*522 BGs (*NM*522) were successfully loaded by simple resuspension of BGs (10 mg BGs were first suspended in PBS, pH 7.4, 0.05 M) with each Sal-complex solutions (Sal-Gd(III), Sal-Mn(II), SalNa, and SalH) in methanol. Ten mg of each Sal-based compound were used for loading *EcN*1917 and 5 mg for *NM*522. The BGs suspensions were stirred for two hours at RT. Subsequently, the loaded BGs were collected by centrifugation at 11 300 g for 15 min, and the pellets were washed three times with Milli-Q^®^ water. One mL aliquots were stored at −20 °C for further analyses.

##### Quantification of Sal-Gd(III)/Sal-Mn(II) Extracted from BGs

In order to determine the amount of Sal-complex within the BGs, 4 mg loaded BGs were resuspended in 0.5 mL of 96% ethanol (Carl Roth, Vienna, Austria), followed by 5 min of ultrasonication. Subsequently, the ethanolic extract was diluted equally with Milli-Q^®^ H_2_O (1:1) and immediately centrifuged at 11,300× *g* for 15 min at 4 °C. FPLC analysis was performed using an AKTÄ Purifier 10 System^®^ (GE Healthcare, Chicago, IL, USA) equipped with a Superdex 10/300 GL column (24 mL; Cytiva, Germany), and UV and conductivity detectors. MeOH-acetate (65:35) was used as an eluent. The quantification was carried out using the peak area method applying free Sal-Mn(II) complex as an external standard.

### 2.3. Physical Measurements

#### 2.3.1. Infrared Spectral Analysis (IR)

The infrared spectra of both compounds were recorded on a Specord-75IR (Carl-Zeiss, Oberkochen, Germany) in a nujol mull.

#### 2.3.2. Electroparamagnetic Resonance Analysis (EPR)

The EPR analysis was performed on a Bruker EMX PremiumX instrument (Karlsruhe, Germany). All measurements were carried out in the X-band at a frequency of microwave electromagnetic radiation 9.45 GHz. Quantitative analysis was performed using Bruker software (Version 1.1b 119) equipped with a spin count option. For temperature variation, a variable temperature unit, ER4141VTM, was used.

#### 2.3.3. Electrospray Ionization Mass Spectrometry (ESI-MS)

ESI-MS spectra were recorded on Waters Micromass ZQ2000 Single Quadrupole Mass spectrometer (Waters, Milford, MA, USA) in MeOH/H_2_O (10% H_2_O), positive mode, in the range of 0–2000 *m*/*z*.

### 2.4. Thermogravimetric Analysis (TGA)

Thermogravimetric analysis (TGA) and differential scanning calorimetry (DSC) was executed using a Netzsch^®^ STA-449 F3 Jupiter instrument from 40 to 800 °C under airflow of 20 mL·min^−1^ as carrier gas with a heating rate of 10 °C min^−1^. Simultaneous thermal analyses allow the measurement of mass changes and thermal effects in the range of 150 °C to 2400 °C. The percentage of mass loss was estimated in the temperature range of 40–800 °C.

### 2.5. In Vitro Relaxivity Measurements

As a reference substance, pure MeOH was used as a negative control, and gadopentetate dimeglumine (Magnevist^®^) and gadoteridol (ProHance^®^) as positive controls. Serial dilutions of BGs loaded with the salinomycin complexes were carried out in ultra-pure Milli-Q^®^ water ranging from 0.02 to 0.1 mM for Mn(II) salinomycinate or Gd(III) salinomycinate, respectively. For the free salinomycinate complexes, six dilutions ranging from 0.01 mM to 3.3 mM (for the Gd(III) salinomycinate) and from 0.02 mM to 1.4 mM (for the Mn(II) salinomycinate) in MeOH with a volume of 0.5 mL were prepared. All probes were measured in Eppendorf safe-lock polypropylene tubes with a diameter of 0.4 cm. The tubes were placed in the center of a plastic box and measured at ambient temperature.

Relaxivity measurements were conducted on a high-field MRI scanner (9.4 Tesla, Bruker Biospec). *T*_1_, *T*_2_, and *T*_2*_ measurements of different contrast media concentrations were performed using the *T*_1_ mapping inversion recovery RARE sequence, the *T*_2_ mapping multislice multiecho spin echo sequence, and the *T*_2*_ mapping multi gradient echo sequence for calculation of relaxivities (*R*_1_, *R*_2_, *R*_2*_).

To determine the *T*_1_ spin-lattice relaxation times, spin echo inversion recovery (IR) sequences with inversion times from 0 to 3500 ms (*TI* = 0, 60, 80, 100, 150, 200, 250, 300, 400, 500, 750, 1000, 1250, 1500, 1750, 2000, 2500, 3500 ms) were used. An adiabatic pulse was applied for B1-insensitive inversion. The other parameters were *TR*/*TE* = 5000/8.1 ms; flip angle 180°; 8 turbo factor 11; FOV 180 × 180 mm; resolution matrix 192 × 192; bandwidth 260 Hz/pixel; and slice thickness 3 mm. For all measurements, the relaxation rates (*R*_1_) and the relaxivities (*r*_1_) were calculated for each substance. The relaxivities (*r*_1_, *r*_2_) were calculated as the slope of the linear regression of *R*_1_ and *R*_2_ as a function of the contrast agent concentration.

### 2.6. Cytotoxicity Tests

#### 2.6.1. Cell Culture

Four adherently growing human cancer cell lines were used for this study: CH1/PA-1 (ovarian teratocarcinoma) cells were a gift from Lloyd R. Kelland (CRC Center for Cancer Therapeutics, Institute of Cancer Research, Sutton, UK), whereas A549 (non-small-cell lung cancer) and SW480 (colon carcinoma) cells were kindly provided by the Institute of Cancer Research, Department of Medicine I, Medical University of Vienna, Austria, and MCF-7 (mammary carcinoma) cells by the Department of Pharmaceutical Sciences, University of Vienna. The first three cell lines were grown in Eagle’s minimal essential medium (MEM) supplemented with L-glutamine (4 mM), sodium pyruvate (1 mM), 1% (*v*/*v*) non-essential amino acid solution, and 10% (*v*/*v*) heat-inactivated fetal calf serum (from BioWest) in 75 cm² flasks at 37 °C under a humidified atmosphere containing 5% CO_2_ in the air. MCF-7 cells were cultured in High Glucose Dulbecco’s Minimal Essential Medium (DMEM) supplemented with 10% (*v*/*v*) fetal calf serum (FCS; from BioWest), L-glutamine (4 mM) and 0.01 mg/mL of human insulin. All cell culture media, supplements, and assay reagents were purchased from Sigma-Aldrich, and all plastic ware from Starlab unless stated otherwise.

#### 2.6.2. MTT Assay

The antiproliferative activity of the compounds was determined by the colorimetric MTT assay (MTT = 3-(4,5-dimethyl-2-thiazolyl)-2,5-diphenyl-2H-tetrazolium bromide). The 1 × 10^3^ CH1/PA-1, 2 × 10^3^ SW480 and 3 × 10^3^ A549 cells were seeded in 100 μL per well into 96-well microculture plates. After 24 h, the tested compounds were dissolved in DMSO (Fisher Scientific), serially diluted in supplemented MEM not to exceed a final DMSO content of 0.5% (*v*/*v*) and added in 100 μL per well. After 96 h, the drug-containing medium was replaced with 100 μL of an RPMI 1640/MTT mixture [six parts of RPMI 1640 medium, supplemented with 10% heat-inactivated fetal bovine serum (FBS), and 4 mM L-glutamine, with one part of MTT solution in phosphate-buffered saline (5 mg/mL)] per well. After incubation for four hours, the MTT-containing medium was replaced with 150 μL DMSO per well to dissolve the formazan product formed by viable cells. Optical density at 550 nm (and at a reference wavelength of 690 nm) was measured with a microplate reader (ELx808, Bio-Tek). The 50% inhibitory concentrations (IC_50_) relative to untreated controls were interpolated from the concentration-effect curves. At least three independent experiments were performed, each in triplicate per concentration level.

#### 2.6.3. Cell Cycle Studies—Impact of Free Sal-H, Sal-Gd(III), and Sal-Mn(II) on the Cell Cycle

Colon carcinoma cells (SW480) and breast adenocarcinoma cells (MCF-7) were seeded in 12-well plates (CytoOne, tissue culture treated) in density of 8 × 10^4^ cells per well in 1 mL of the corresponding medium. After a recovery time of 24 h, cells were treated with different concentrations of Sal-H, Sal-Mn(II), and Sal-Gd(III). For this purpose, the test substances were dissolved in DMSO and diluted in a medium such that the maximum concentration of DMSO in the cells did not exceed 0.5%. For assay validation, the well-known cell cycle inhibitors etoposide and gemcitabine were applied as positive controls. Plates were incubated with test compounds for 24 h at 37 °C, 5% CO_2_. Following the exposure, the medium was completely removed, and adherent cells were gently washed twice with ice-cold PBS. Propidium iodide (PI, 1.0 mg/mL) and HFS-buffer (0.1% (*v*/*v*) Triton X-100, 0.1% (*w*/*v*) sodium citrate in Milli-Q^®^ water) were mixed for staining to yield a final PI concentration of 40 μg/mL. The ice-cold staining solution was added to the cells (500 μL/well). The staining was carried out overnight at 4 °C in the dark. To prepare the samples for measurement, the staining solution was vigorously pipetted against the surface to achieve sufficient resuspension of the cells. For each sample, 200 μL of the stained cell suspensions were added to a 96-well round-bottom FACS plate (Falcon^®^). The fluorescence of all samples stained with a DNA-intercalating agent was measured no longer than 24 h after staining using a flow cytometer (Guava easyCyte 8HT, Millipore^®^). GuavaSoft™ software was used in the InCyte-modus, and 10,000 events/probes were counted.

The flow cytometry data sets were evaluated in FlowJo software (v 10.8.1). The single-cell populations were gated based on forward and side scatters characteristics. The Watson Pragmatic model was applied as a standard method to analyze the resulting histograms: the recorded red fluorescence intensities of cells in the S phase were located between normally distributed G1/G0 and G2/M peaks. The number of cells in each phase was calculated from the model by means of integration. Means and standard deviations were calculated from at least three independently performed experiments. The figures were adjusted in GIMP freeware (v 2.10.24).

## 3. Results

### 3.1. Synthesis and Characterization

The elemental analysis demonstrated that salinomycin sodium reacts with Gd(III) and Mn(II) to form homometallic mononuclear complexes of composition [Gd(C_42_H_69_O_11_)_3_(H_2_O)_3_] and [Mn(C_42_H_69_O_11_)_2_(H_2_O)_2_] respectively. The ESI-MS spectra of both complexes corresponded to the elemental analysis and contained several signals due to the dissociation of water molecules, ligand anions, and/or complexation with Na^+^ (ESI-MS Spectra, Appendix A).

In the IR spectrum of the Gd(III) salinomycinate, two characteristic bands at 1540 cm^−1^ and 1400 cm^−1^ corroborated the monodentate coordination mode of the carboxylate anion to Gd(III). The shift of the band for the carbonyl group from 1700 cm^−1^ to 1690 cm^−1^ compared to the IR spectrum of salinomycinic acid proved the participation of the carbonyl group of the antibiotic in weak hydrogen bonds. The strong, broadband at 3400 cm^−1^ confirmed the presence of hydrogen-bonded hydroxyl groups (IR Spectra, Appendix A).

The IR spectrum of Mn(II) salinomycinate consisted of four characteristic bands at 1400 cm^−1^, 1550 cm^−1^, 1690 cm^−1^, and 3400 cm^−1^. The first two bands were assigned to symmetric and asymmetric stretching vibrations of the deprotonated carboxyl group. The difference between both bands (Δν ≤ 150 cm^−1^) [21] confirmed the monodentate coordination of the deprotonated carboxyl group to the paramagnetic metal center. Similar to the IR spectrum of Gd(III) salinomycinate, the shift of the band for the carbonyl group from 1700 cm^−1^ to 1690 cm^−1^, compared to the IR spectrum of salinomycinic acid, confirmed the participation of the carbonyl group of the antibiotic in weak hydrogen bonds. The broad band at 3400 cm^−1^ was assigned to the stretching vibrations of hydrogen-bonded hydroxyl groups.

The EPR spectra of the Gd(III) salinomycinate, recorded at 100 K and 295 K, are shown in Figure 1. A series of signals, distributed from 0 to 500 mT, were registered. The spectra of Figure 1 demonstrated that, in the whole temperature range, the spectrum remained unchanged independent of the number and the positions of the signals, and the temperature lowering led to an increase in signal intensities. Upon closer observation, some prominent features of the spectra were revealed—the signals with g_eff_ ≈ 6.0, 2.8, and 2.0. This set of signals is known as the U (ubiquitous) spectrum of Gd^3+^ ions. It should be mentioned that a characteristic feature of the U-spectrum is the almost equal intensity of the signals with geff ≈ 6.0, 2.8 [25]. The regarded complex spectrum with the above-described arrangement of the EPR signals is due to isolated Gd^3+^ ions in a low symmetry field with the zero-field splitting parameter D > 0.3 cm^−1^. Under such a geometry, for the Gd^3+^ ions (^8^S_7/2_, 4f^7^ electronic configuration) a multitude of electron transitions is possible between eight energy levels (m_s_ = ±1/2, m_s_ = ±3/2, m_s_ = ±5/2 and m_s_ = ±7/2). It is noteworthy that the considered U-spectrum is assigned to Gd^3+^ ions with a coordination number higher than six in non-crystalline materials [26].

The Mn(II) salinomycinate complex was studied by EPR analysis at 100 and 295 K (Figure 2). At both measurement temperatures, identical spectrum features were observed. In the central magnetic field region, a relatively broad signal with the following EPR parameters was registered: g_eff_ ≈ 1.99, ∆H_pp_ ≈ 54.0 mT. Six clearly distinct narrow lines are superimposed on it (Figure 2, inset) at a distance of approximately 9.5 mT. At a lower magnetic field, an additional signal could be noticed with a g-factor around 5.0. The signal with g ≈ 1.99, as well as the six narrower lines located on it, were assigned to Mn^2+^ ions, which are characterized by ground state ^6^S_5/2_ and electron and nuclear spin number—S = 5/2, I = 5/2. The registered sextet of lines was attributed to the hyperfine structure of Mn^2+^ ions. Principally, the hyperfine structure possessed a unique value of a hyperfine splitting constant, A_hfs_, depending on the closest surrounding Mn^2+^ ions. The experimentally determined constant (A_hfs_) for Mn(II) salinomycinate in this study was 9.5 mT. For comparison, the hyperfine splitting constant of the aqua complex [Mn(H_2_O)_6_]^2+^ was known to be 9.4 mT and, thus, approximated the corresponding constant found for the Mn(II) salinomycinate complex [27]. This similarity implies the coordination of Mn^2+^ to water molecules and OH groups in the studied complex.

The registration of the EPR signal with g ≈ 5 suggests the presence of a large zero-field splitting constant, D > 0.3 cm^−1^. Therefore, the EPR spectrum of the Mn(II) salinomycinate complex showed the presence of isolated Mn^2+^ ions in a low-symmetry ligand field.

### 3.2. Thermogravimetric Analysis (TGA)

TGA and DSC analysis of the two Sal metal complexes were studied. The heating rates were suitably controlled at 10 °C min^−1^ under airflow of 20 mL·min^−1,^ and the weight loss was measured from ambient temperature to 800 °C, depicted in Figure 3.

The two Sal metal complexes were thermally stable up to 55 °C (TGA; Figure 3A,B). A mass loss in percentage was observed with a broad and flat endothermic peak at 100 °C in the DSC curve for Sal-Mn(II) (Figure 3A) and a more defined endothermic peak at 70 °C for Sal-Gd(III) (Figure 3B). This may be due to the removal of coordinated and absorbed water as depicted in the light green color (3% for the Sal-Mn(II) complex, Figure 3A and 5% for the Sal-Gd(III) one, Figure 3B). The second drop in the masses starting from 150 °C to 475 °C (Figure 3A) and from 150 °C to 440 °C (Figure 3B), highlighted in dark green, can be assigned to the two-step decomposition of the bis- or trisalinomycinates (corresponding to 73% and 72% mass loss respectively (Figure 3A,B). This thermal process is associated with two small exothermic peaks for each Sal complex at 190 °C and 370 °C and at 260 °C and 380 °C in the DSC curve (Figure 3A,B; red-orange line on the right). The final and last step, with a mass loss of 15% between 480–540 °C for the Sal-Mn(II) and 450–540 °C for the Sal-Gd(III) compound, correlated with a strong exothermic signal at 500 °C in the DSC curve. This was most probably due to the oxidation and combustion of the rest of organic matter and the formation of MnO or Gd_2_O_3_ as final residues. All these findings are in good correlation with published characterizations of metal complexes using TGA-DSC analyses under air atmosphere [28,29].

### 3.3. In Vitro MRI

The magnetic susceptibility of the free Sal-complexes and the loaded ones into BGs was determined in a dilution series in MeOH or in Milli-Q^®^ water, respectively, and compared to two clinically used contrast agents, the linear gadopentetate dimeglumine (Magnevist^®^) and cyclic gadoteridol (ProHance^®^). A steep signal increase with increasing concentrations has been observed for the prepared contrast agents (Figure 4).

## 4. Cell Culture Studies

### 4.1. Cytotoxicity Studies

The cytotoxic activity of the salinomycinate complexes alone and when loaded into *EcN1917* and *NM522* was tested against three human cell lines and showed pronounced anticancer potency (Figure 5 and Figure 6 and Table 1). Our results for the cytotoxicity of the paramagnetic complexes of Sal with Mn(II) and Gd(III), and loaded into BGs were compared with those for the cytotoxic activity of four Pt-containing conventional chemotherapeutic drugs and are summarized in Table 3 [30,31]. Data for the cytotoxicity of salinomycin (SalH) and salinomycin-sodium (SalNa) are also given. The results revealed that both paramagnetic complexes of salinomycin are more cytotoxic against the tested tumor cell lines compared to salinomycin and its sodium complex. The effect was more pronounced on SW480 and CH1/PA-1 cell lines. Both paramagnetic complexes of salinomycin showed dose-dependent cytotoxicity against the tested tumor cell lines (Figure 5 and Figure 6).

After incorporation into empty bacterial envelopes, both Sal complexes retained their antitumor activity. Notably, both *NM522* formulations loaded with half the amount of the Sal complex as that for the *EcN1917* exhibited similar cytotoxic efficacy at lower concentrations (Figure 7 and Figure 8). The non-loaded *EcN1917* and *NM522* show no activity.

Surprisingly, some of the loaded BGs exerted higher cytotoxic efficacy than the free Sal-Mn(II) and Sal-Gd(III). In particular, *EcN 1917* loaded with Sal-Gd(III) possessed an IC_50_ of only 0.09 μM against A549 and 0.12 μM for *Ec NM522* loaded with half the starting Sal complex as that for *EcN 1917*. The cytotoxic effect of both loaded BGs on SW480 was also more pronounced, with an IC_50_ of 0.27 μM and 0.40 μM relative to the 0.36 μM of the Sal-Gd(III) alone. Tumor cell eradication of CH1/PA-1 cells was similar after the application of both free and loaded Sal-Gd(III). Control Sal-Na exerted less cytotoxic activity after entrapment into BGs. For both A549 and SW480, cancer cell line cytotoxicity decreased after loading of Sal-Mn(II), with IC_50_ values of 0.33 and 0.75 μM, while the effect remained unchanged for CH1/PA-1. Altogether, the prepared free and loaded compounds possessed a superior cytotoxic efficacy on all three cancer cell lines tested (Table 1).

### 4.2. Cell Cycle Studies

In general, the SW480 cell line was more sensitive to tested compounds than the MCF-7 cell line. The concentrations of 4–8 μM were sufficient for altering the cell cycle distribution in colon cancer cells upon 24 h treatment (Appendix A). Salinomycinic acid (Appendix A) and its metal complexes (Appendix A) diminished the SW480 cell fraction in the G0/G1-phase (by 8–17%) and correspondingly increased the cell fraction in the S-phase (by 9–10%), followed by a G2/M-phase increase at the highest concentrations (by 7–8%). Effects on the cell cycle distribution of MCF-7 breast cancer cells were more pronounced at higher (16–32 μM) concentrations (Appendix A). The tested compounds mainly decreased the number of cells in the G0/G1-phase (by 15–18%) and increased the number of cells in the G2/M-phase (5–13%), with a minor effect on the S-phase fraction (±6%).

The concentration-dependent effect of the individual compounds on SW480 and MCF-7 cells is exemplified in Appendix A, respectively. Sal-H was the least active compound in both cell lines, with moderate activity at the highest concentrations applied. Sal-Mn(II) and Sal-Gd(III) demonstrated a more pronounced effect on the cell cycle in both colon and breast cancer cells. The direct correlation between the number of salinomycin ligands in the complex and the potency to induce cell cycle perturbations could be observed. Sal-Gd(III) demonstrated the highest activity in both cell lines (Appendix A).

## 5. Discussion

Pilot clinical trials have demonstrated that salinomycin inhibited the disease progression in patients diagnosed with invasive carcinoma. Data revealed antibiotic-induced apoptosis in the metastatic cancer cells, and no severe, long-lasting side effects were observed [32]. These promising results demonstrated the remarkable potency of salinomycin for cancer treatment. Coordination of the antibiotic to divalent metal ions can further enhance its cytotoxicity and diminish its toxicity [21,32,33]. Chelated metal ions can also provide a convenient handle for bioconjugation with other molecules via axial coordination. Herein, we explored the structures of two paramagnetic complexes of salinomycin and demonstrated, for the first time, their potency as theranostic probes. Moreover, the loading of the novel potent theranostics into BGs may provide a targeted drug delivery that can circumvent the water insolubility of the free Sal-complexes and pave the way to clinical translation.

The IR spectra of salinomycin with Mn(II) and Gd(III) reported in this study were similar to the IR spectra of [Co(C_42_H_69_O_11_)_2_(H_2_O)_2_], [Ni(C_42_H_69_O_11_)_2_(H_2_O)_2_], [Cu(C_42_H_69_O_11_)_2_(H_2_O)_2_], [Zn(C_42_H_69_O_11_)_2_(H_2_O)_2_], and [Cd(C_42_H_69_O_11_)_2_(H_2_O)_2_] discussed previously [33,34]. The literature spectroscopic data for M(II) salinomycinates have suggested that the organic ligand was coordinated to the metal center via a terminal deprotonated carboxyl group and a terminal secondary hydroxyl group [21,34]. A distorted octahedral molecular geometry of the complexes of salinomycin with M(II) has been proposed [21,34].

Herein, we applied a solid-state EPR spectroscopy to obtain more detailed information about the structures of salinomycin complexes with Gd(III) and Mn(II). The complex EPR spectrum of Gd(III) salinomycinate is due to Gd^3+^ ions placed in a crystal field with low symmetry and characterized by a high splitting constant at zero magnetic field [35,36]. For the Mn(II) salinomycinate, the described broad signal consists of six superimposed lines, which are attributable to the ultrafine interaction characteristics of Mn(II) ions (nuclear spin I = 5/2). The calculated superfine interaction constant A_hfs_ is ca. 9.5 mT. For comparison, the superfine interaction constant for [Mn(H_2_O)_6_]^2+^ is 9.4 mT [27]. The additional signal at g ≈ 5.0 suggests that the cleavage constant of Cramer doublets in a zero magnetic field is significant (above 0.3 cm^−1^). A quantitative EPR analysis was additionally applied for the estimation of the metal ion to ligand ratio. The ratio Gd^3+^ to salinomycin was established to be 1:5. The discrepancy to the formula obtained via elemental analysis could be explained by some loss of EPR signal as a result of the large zero-field splittings constant value. The experimentally evaluated ratio Mn^2+^ to salinomycin was found to be 1:2.5 showing a good agreement with the results from elemental analysis. The presence of aqua ligands in the coordination sphere of the Sal-Mn(II) complex was also supposed by the EPR analysis.

When taken together, the data from the elemental analysis and spectroscopic studies (IR, ESI-MS, EPR) allowed us to conclude that the complexes of salinomycin with Gd(III) and Mn(II) possess a tricapped, trigonal, prismatic geometry (Figure 9) and an octahedral molecular geometry (Figure 10) respectively. The salinomycinate monoanions are coordinated to the metal center via a deprotonated carboxyl group and a terminal secondary hydroxyl group. The water molecules coordinated to the metal center stabilize the pseudocyclization of the complex by hydrogen bonds with the organic ligand. In addition, TGA and DCS studies confirmed coordinated water molecules for both Gd(III) and Mn(II).

Relaxivity (*r*_1_ or *r*_2_) in MRI is defined as the paramagnetic enhancement of the water protons relaxation rates (*R*_1_ and *R*_2_) caused by 1 mM contrast agent concentration. It thus directly relates to the MRI efficiency of the contrast agent. The majority of CAs are based on paramagnetic ions (predominately Gd(III)) chelated in stable complexes, which reduce the longitudinal relaxation time (*T*_1_) of water protons in the body. Consequently, positive (bright) *T*_1_-weighted MR images can be seen (*T*_1_-enhancers).

As new MRI contrast agents, both the free and the loaded into BGs Gd(III) salinomycinate and Mn(II) salinomycinate were evaluated for their ability to modify the *R*_1_ of water protons on a high-field scanner (9.4 Tesla; BioSpec 94/30USR, Bruker, Germany). The measured relaxivity values (*r*_1_) of Gd(III) salinomycinate and Mn(II) salinomycinate, 5.4 and 2.5 1/mM*s, respectively, are higher than or as high as the most clinically applied probes (Figure 4). Even though the *r*_1_ of Mn(II) salinomycinate is not as high as that of the salinomycin complex with Gd(III), Sal-Mn(II) causes an excellent contrast in MRI and may serve as a good alternative to the Gd-based contrast agents (GBCA). Moreover, Mn-based agents have shown a better diagnostic performance than Gd-based agents in certain disease areas, such as pancreatic lesions [37]. Compared to other Mn-based small molecular MRI contrast agents, Mn(II) salinomycinate (*r*_1_ = 2.5 1/mM*s) shows a relaxivity as high as that recently reported by Wang et al. for Mn-PyC3A-3-OBn (*r*_1_ = 2.6 1/mM·s, in Tris buffer at 1.4 T), a novel Mn(II)-based complex designed for liver-specific imaging [38]. The only Mn(II)-based complex that was clinically available for some time was the liver-specific Mangafodipir^®^ or Teslascan^®^ ([Mn(dpdp)]^4−^), in which the relaxation enhancement arises from the Mn(II) ion released from the complex in vivo [39]. The Mn(II) complex of salinomycin, described in this study, exerts a higher relaxivity than Teslascan^®^ generates in vivo (*r*_1_ = 2.3 1/mM·s, at 1 T, MRI-relaxivity MR-TIP: Database) and a comparable strong signal increase compared to the aqueous solutions of Teslascan^®^ (*r*_1_ = 2.8 1/mM·s) [39].

All Sal-complexes loaded into BGs showed a superior signal enhancement in MRI, with *r*_1_ = 2.578 1/mM·s and a pronounced *r*_2_ = 36.74 1/mM·s for the bacterial cells, carrying Sal-Mn and an increased *r*_1_ = 6.674 1/mM·s for those with entrapped Sal-Gd(III). This is comparable with the *r*_1_ of clinically used Magnevist^®^ and ProHance^®^ and about ten times higher when compared to the *r*_2_ = 3.70 [1/mM·s] reported for Mangafodipir^®^. The contrast efficiency of the loaded BGs with Sal-Gd(III) (*r*_1_ = 6.674 1/mM·s) improved approximately 1.8 times compared to the clinically applied Magnevist^®^ (*r*_1_ = 3.76 [1/mM·s]).

The summarized proton relaxivities of our Gd(III) and Mn(II) complexes of salinomycin with regard to various published low-molecular-weight paramagnetic complexes ere shown in Table 2.

With regard to the *T*_1_ values and the calculated relaxivities (*r*_1_), it is known that, at higher field strength, the relaxivity values tend to decrease slightly [44]. Nevertheless, the table above truly illustrates the great potential of the newly prepared salinomycinates as contrast agents for MRI. In addition, *r*_1_ is always influenced by several physical and chemical parameters, such as the strength and homogeneity of the applied magnetic field, the hydration state of the contrast agent, its molecular size, internal and anisotropic rotation, water exchange rates, and molecular tumbling [45].

Notably, the chemical composition of Gd(III) salinomycinate and Mn(II) salinomycinate comprises three and two inner-sphere water molecules, respectively, which presumably contributes to higher relaxivity values and, in turn, to an increased contrast in MRI. It is known that the interaction dynamics between water and Gd(III) or Mn(II) complexes highly affect the relaxivity, while the mechanism remains unclear. Based on Solomon-Bloembergen-Morgan (SBM) theory, the direct water coordination to the paramagnetic metal centers is the major contributor to *T*_1_ inner-sphere relaxivity [40].

As expected, the signal enhancement in MRI caused by the Sal-complexes loaded within the bacterial cells was superior compared to the non-loaded free Sal-Gd(III) and Sal-Mn(II), even at lower Gd(III) or Mn(II) concentrations (0.06–0.1 mM Gd(III) and 0.05–0.1 mM Mn(II) applied). This fact could be explained by a slowed molecular tumbling of the macromolecular constructs and a more effective water exchange rate provided by the aqueous media in which the BGs were resuspended.

Remarkably, the complexes of salinomycin with Gd(III) and Mn(II) and the loaded BGs exerted superior cytotoxicity on A549 compared to all conventional Pt-containing clinical chemotherapeutics (Table 3) [30,31]. All compounds demonstrated much higher antitumor activity against SW480 than did satraplatin, cisplatin, and carboplatin, and comparable cytotoxicity to oxaliplatin [30,31]. The cytotoxicity of salinomycin with Mn(II) against CH1/PA-1 was similar to that of oxaliplatin and satraplatin, more pronounced in contrast to carboplatin, and lower compared to cisplatin. The cytotoxicity of the Gd(III) complex of salinomycin against CH1/PA-1 was either much more elevated set against clinical chemotherapeutics (carboplatin, oxaliplatin) or comparable (satraplatin, cisplatin) [30,31].

The antiproliferative activity of Mn(II) salinomycinate against the more chemo-resistant A549 cell line was much more pronounced compared to that of Mn(II) with other ligands [47,48,49,50,51]. The IC_50_ value for [Mn(Sal)_2_(H_2_O)_2_] was from eight to 984 times lower compared to IC_50_ values for the other Mn(II) complexes. This enormous difference in the cytotoxicity of Mn(II) complexes against the A549 cell line could not be attributed to the differences in the experimental protocols. We found only one study where the cytotoxicity data for Gd(III) complexes with other ligands against A549 were reported [52]. The cytotoxicity of the Sal-Gd(III) complex against the A549 cell line was superior compared to the Gd(III) complex described in the literature (the IC_50_ value was 733 times lower compared to the value for the published complex) [53].

The results demonstrate that salinomycinic acid and its metal-based complexes Sal-Gd(III) and Sal-Mn(II) may induce cell cycle perturbations in both breast cancer MCF-7 and colon cancer SW480 cells, shifting the cell number distribution from G1/G0- towards the S- and G2/M-phases. In line with our data, Niwa et al. reported that a 24 h treatment of MCF-7 cells with 20 μM salinomycin significantly increased cell numbers in the G2/M-phase at the expense of cells in the G0/G1-phase [54].

The combination of MRI contrast agents and a highly effective anticancer drug into a single construct, such as a Sal-based theranostic complex formulated within bacterial ghosts as a transport and targeting device, may offer a powerful “all in one” tool for diagnosis, therapy, and monitoring of various cancers.

## 6. Conclusions

In this study, we synthesized and characterized, for the first time, the Gd(III) complex of salinomycin. We provide new data about the structure and cytotoxicity of Mn(II) salinomycinate. Further loading of both Sal-complexes into *EcN*1917 and *NM*522 BGs as transport vehicles was successfully carried out. The higher payload of paramagnetic metal ions and an anticancer drug within the BGs may provide better biocompatibility, as well as low/no toxicity or undesired side effects, making these new formulations suitable for clinical applications. To the best of our knowledge, this is the first study that demonstrates the potential use of salinomycin complexes with Gd(III) and Mn(II) as theranostic agents for MRI. This is the first report that a natural, polyether ionophorous antibiotic can act as a chelator of paramagnetic metals and an anticancer drug, exerting a superior cytotoxic effect and favorable MRI properties.

## Figures and Tables

**Figure 1 pharmaceutics-14-02319-f001:**
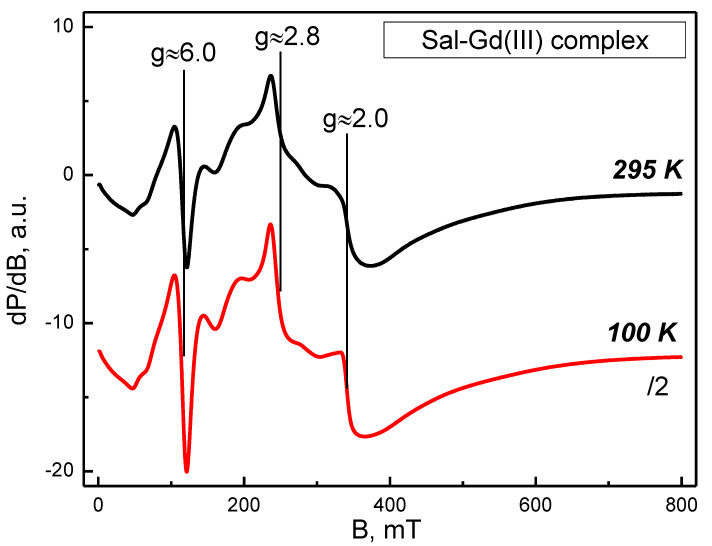
EPR spectra of Sal-Gd(III) complex (solid-state) at 295 and 100 K. The U-spectra signals (g ≈ 6.0, g ≈ 2.8, and g ≈ 2.0) are designated.

**Figure 2 pharmaceutics-14-02319-f002:**
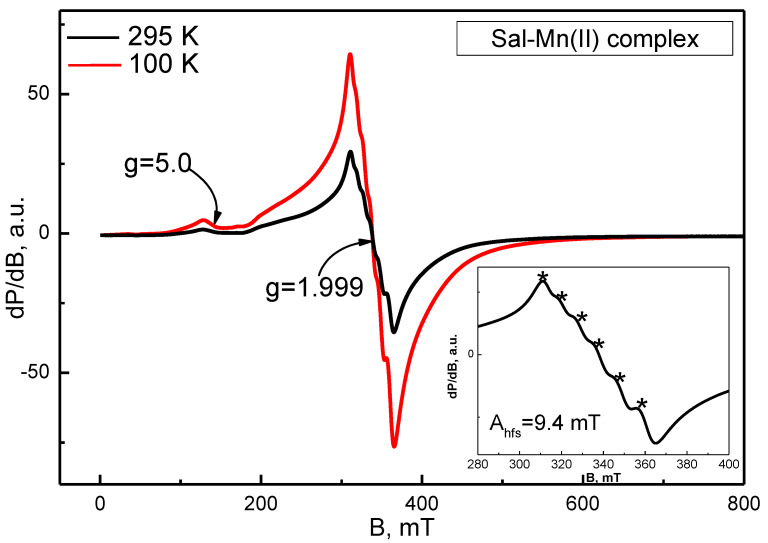
EPR spectra of Sal-Mn(II) complex at 295 and 100 K. The six lines of the Mn^2+^ hyperfine structure are shown in the inset.

**Figure 3 pharmaceutics-14-02319-f003:**
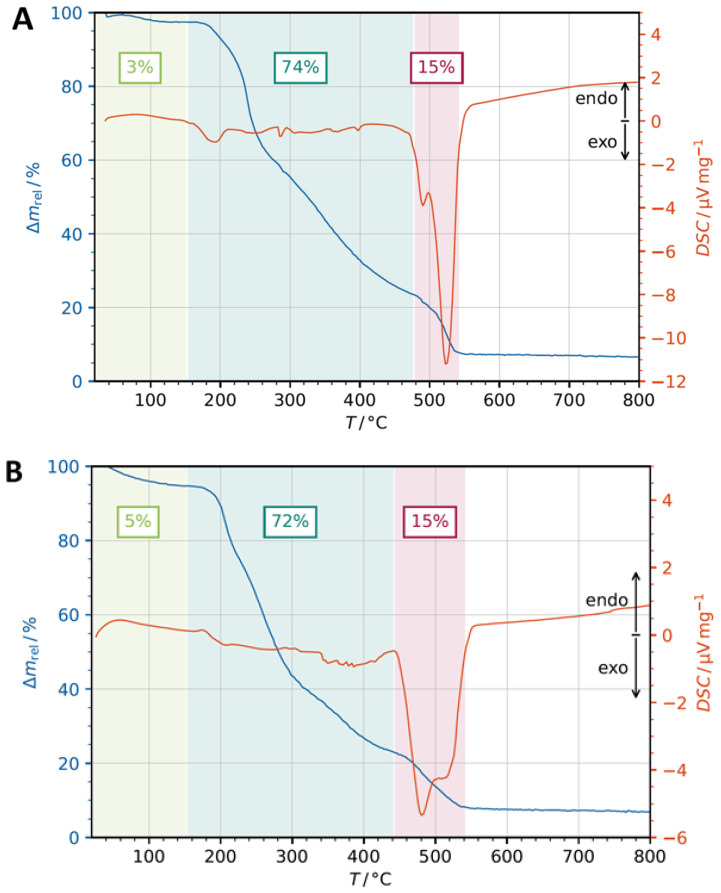
DSC and mass loss profiles of (**A**) Sal-Mn(II) and (**B**) Sal-Gd(III) complexes.

**Figure 4 pharmaceutics-14-02319-f004:**
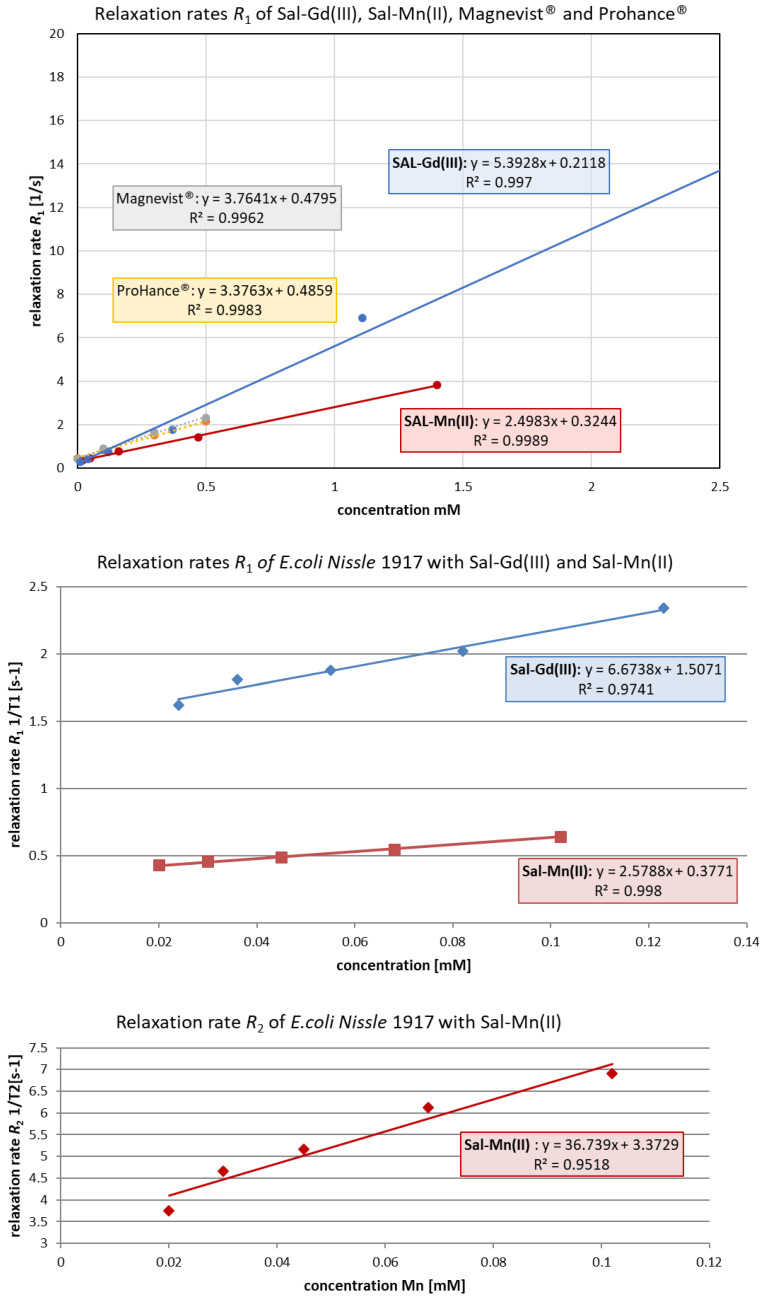
Plots of relaxation rates (*R*_1_/*R*_2_, [1/s]) to concentration (mM) curves for Sal-Gd(III), Sal-Mn(II), and *EcN1917*-BGs loaded with both Sal complexes compared to clinically applied gadopentetate dimeglumine (Magnevist^®^, linear, grey line) and gadoteridol (ProHance^®^, cyclic, yellow line). For better comparison, one point from the Sal-Gd(III) graph corresponding to 0.33 mM concentration has been omitted. A strong signal increase in MRI using either the non-loaded Sal-Gd(III) and Sal-Mn(II) or the loaded complexes was shown, comparable with or superior to the effect caused by the clinically most applied CAs, gadopentetate dimeglumine and gadoteridol. The Sal-Gd(III) compound caused a predominant *T*_1_ effect only; thus, just the *R*_1_ dependence with increasing concentration is shown.

**Figure 5 pharmaceutics-14-02319-f005:**
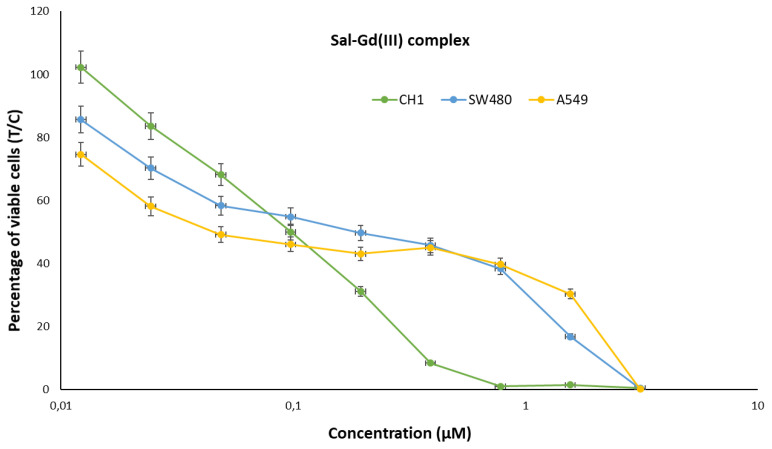
Concentration-effect curves of the Sal-Gd(III) complex in A549 (yellow), CH1/PA-1 (green), and SW480 (blue) cells, determined by the MTT assay after 96 h treatment. Values are normalized relative to untreated controls and represent means and standard deviations of at least three independent experiments.

**Figure 6 pharmaceutics-14-02319-f006:**
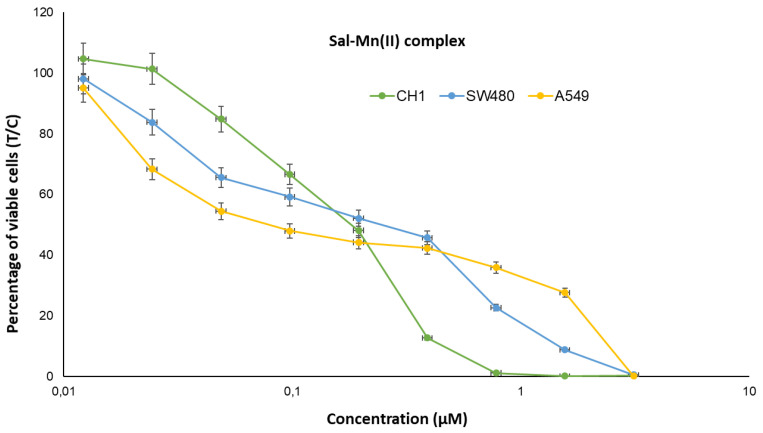
Concentration-effect curves of the Sal-Mn(II) complex in A549 (yellow), CH1/PA-1 (green), and SW480 (blue) cells, determined by the MTT assay after 96 h treatment. Values are normalized relative to untreated controls and represent means and standard deviations of at least three independent experiments.

**Figure 7 pharmaceutics-14-02319-f007:**
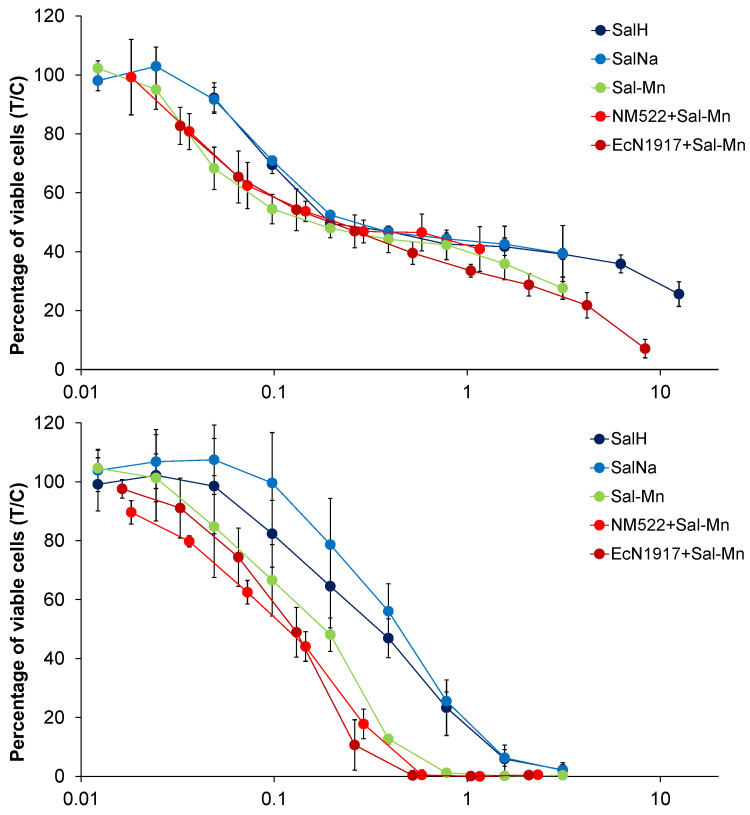
Concentration-effect curves of Sal-H, Sal-Na, Sal-Mn(II), *NM522* + Sal-Mn(II), and *EcN1917* + Sal-Mn(II) in A549 (top), CH1/PA-1 (middle), and SW480 (bottom) cells, determined by the MTT assay after 96 h treatment. Values are normalized relative to untreated controls and represent means and standard deviations of at least three independent experiments.

**Figure 8 pharmaceutics-14-02319-f008:**
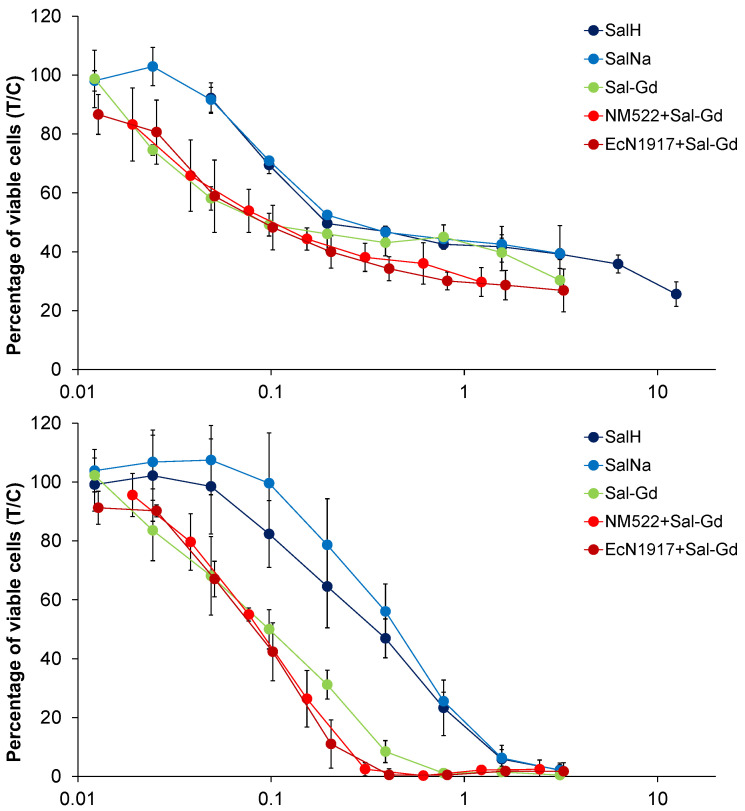
Concentration-effect curves of SalH, SalNa, Sal-Gd(III), *NM522* + Sal-Gd(III) and *EcN1917* + Sal-Gd(III) in A549 (top), CH1/PA-1 (middle) and SW480 (bottom) cells, determined by the MTT assay after 96 h treatment. Values are normalized relative to untreated controls and represent means and standard deviations of at least three independent experiments.

**Figure 9 pharmaceutics-14-02319-f009:**
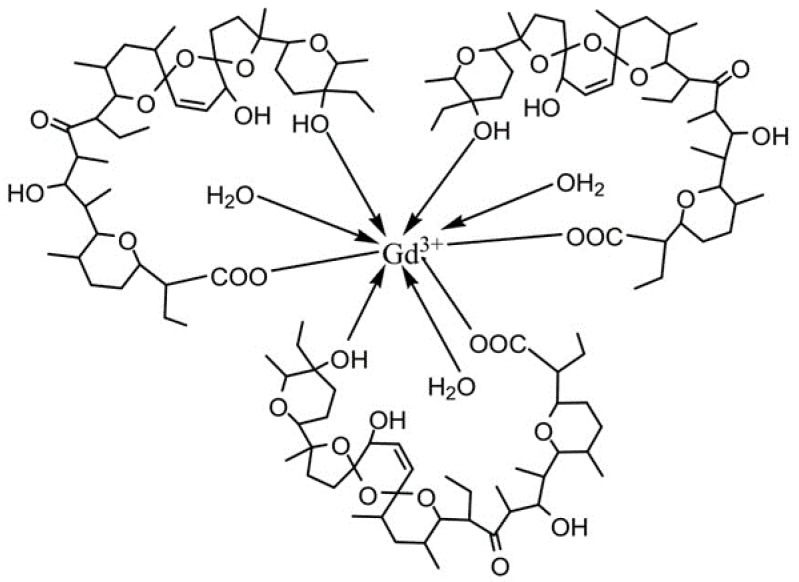
Proposed chemical structure of [Gd(C_42_H_69_O_11_)_3_(H_2_O)_3_].

**Figure 10 pharmaceutics-14-02319-f010:**
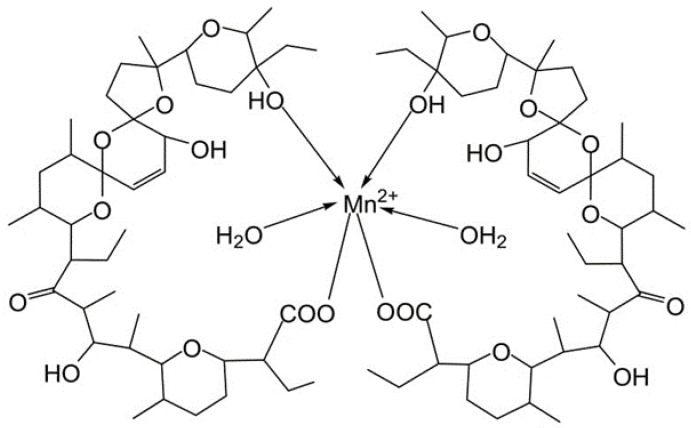
Proposed chemical structure of [Mn(C_42_H_69_O_11_)_2_(H_2_O)_2_].

**Table 1 pharmaceutics-14-02319-t001:** Cytotoxicity evaluation of SalH, SalNa, free Gd(III) and Mn(II) salinomycinates, and with loaded BGs: mean IC_50_ values (in μM) ± standard deviations from at least three independent MTT assays in each of three human cancer cell lines.

		Cell Line	A549	SW480	CH1/PA-1
	Sample				
1.	Sal-H		0.23 ± 0.06	1.1 ± 0.6	0.32 ± 0.12
2.	Sal-Na		0.27 ± 0.02	0.88 ± 0.44	0.43 ± 0.11
3.	[Mn(Sal)_2_(H_2_O)_2_] = Sal-Mn(II)		0.19 ± 0.11 *	0.52 ± 0.22	0.17 ± 0.05
4.	[Gd(Sal)_3_(H_2_O)_3_] = Sal-Gd(III)		0.15 ± 0.12 *	0.36 ± 0.12	0.093 ± 0.025
5.	EcN1917 + 10 mg/mL Sal-Mn(II)		0.22 ± 0.09	0.55 ± 0.06	0.12 ± 0.03
6.	EcN1917 + 10 mg/mL Sal-Gd(III)		0.093 ± 0.043	0.28 ± 0.14	0.086 ± 0.021
7.	NM522 + 5 mg/mL Sal-Mn(II)		0.28 ± 0.17 *	0.54 ± 0.27	0.12 ± 0.02
8.	NM522 + 5 mg/mL Sal-Gd(III)		0.092 ± 0.041	0.31 ± 0.15	0.088 ± 0.008

* The bigger standard deviation is due to a flatter curve.

**Table 2 pharmaceutics-14-02319-t002:** Comparison of proton relaxivity *r*_1_ in [1/mM·s] for Sal-Gd(III), Sal-Mn(II), and *E. coli Nissle 1917* loaded with the two Sal complexes with various low-molecular-weight Gd(III) and Mn(II) complexes.

Complex	Relaxivity Values *r*_1_/*r*_2_, [1/mM·s] (RT/20 °C/37 °C)	Magnetic Field Strength, [T]
Gd-DTPA^2−^ (Magnevist^®^)	3.76/n.d.	9.4
Gd-HPDO3A (ProHance^®^)	3.37/n.d.	9.4
JTIP1/[Mn(Sal)_2_(H_2_O)_2_]	2.50/n.d.	9.4
JTIP3/[Gd(Sal)_3_(H_2_O)_3_]	5.40/n.d.	9.4
*E. coli Nissle 1917* loaded with JTIP1/[Mn(Sal)_2_(H_2_O)_2_]	2.58/36.74	9.4
*E. coli Nissle 1917* loaded with JTIP3/[Gd(Sal)_3_(H_2_O)_3_]	6.67/n.a.	9.4
[Mn(DPDP)]^4−^ (Teslascan^®^) [39]	2.80/3.70	4.7
Mn-PyC3A [40]	3.4/n.d.	3
Mn-EDTA-BTA [41]	3.50	0.47
Mn(HBET-NO_2_)]^2−^ [42]	2.33	1.4
Mn-CDTA^2−^ [43]	2.99	0.47
Mn-EDTA^2−^ [43]	2.12	0.47
Mn-DTPA^3−^ [43]	1.57	0.47

**Table 3 pharmaceutics-14-02319-t003:** Cytotoxic activity of paramagnetic complexes of salinomycin with Gd(III) and Mn(II), loaded BGs and Pt-containing clinical chemotherapeutics: mean IC_50_ values (in μM) ± standard deviations from at least three independent MTT assays in each of three human cancer cell lines.

		Cell Line	A549	SW480	CH1/PA-1
	Sample				
1.	Sal-H		0.23 ± 0.06	1.06 ± 0.58	0.32 ± 0.12
2.	Sal-Na		0.27 ± 0.02	0.88 ± 0.44	0.43 ± 0.11
3.	[Mn(Sal)_2_(H_2_O)_2_] = Sal-Mn(II)		0.19 ± 0.11	0.52 ± 0.22	0.17 ± 0.05
4.	[Gd(Sal)_3_(H_2_O)_3_] = Sal-Gd(III)		0.15 ± 0.12	0.36 ± 0.12	0.09 ± 0.03
5.	*EcN1917* + 10 mg/mL Sal-Mn(II)		0.33 ± 0.14	0.82 ± 0.10	0.19 ± 0.04
6.	*EcN1917* + 10 mg/mL Sal-Gd(III)		0.09 ± 0.04	0.27 ± 0.13	0.082 ± 0.020
7.	*NM522* + 5 mg/mL Sal-Mn(II)		0.75 ± 0.46	1.44 ± 0.74	0.32 ± 0.054
8.	*NM522* + 5 mg/mL Sal-Gd(III)		0.12 ± 0.05	0.40 ± 0.19	0.112 ± 0.010
Satraplatin [30]	Pt(NH_3_)(cha)Cl_2_(OAc)_2_		6.4 ± 0.4	1.5 ± 0.1	0.10 ± 0.12
Cisplatin [30]	Pt(NH_3_)_2_Cl_2_		6.2 ± 1.2	3.3 ± 0.2	0.077 ± 0.006
Carboplatin [46]	Pt(NH_3_)_2_(CBDCA)		91 ± 102	37 ± 1	0.81 ± 0.17
Oxaliplatin [31]	Pt(DACH)(ox)		0.98 ± 0.21	0.29 ± 0.05	0.18 ± 0.01

## Data Availability

All other relevant data of this study are available from the corresponding authors upon reasonable request.

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
