# Peer review of "Novel Salinomycin-Based Paramagnetic Complexes—First Evaluation of Their Potential Theranostic Properties"

_pharmaceutics, 2022, doi:10.3390/pharmaceutics14112319_

Round 1

Reviewer 1 Report

The paper is well described and the results are commented in relatively detail. The conclusion supports the experimental data presented and the data looks quite promising for the theranostic field.

There are just small changes to the English and style of the manuscript to be made by the Authors, e.g .:

1) "in vitro" and "in vivo" must be written in italics;

2) Drugs names such as Magnevist... should be written with the copyright symbol "(R)" at the end of the word (in superscript). The same goes for ultrapure water Milli-Q .

3) Check the "numbers" in the chemical formulas: many are not written in subscript. Same goes for the charge state of the ions (in superscript) (see page 2 for example).

4) The parameters "T1, T2, T2 *, r1" should be written in italics (the letter) and in subscript (the number);

5) The ESI-MS method is not of the spectroscopic type but it is a spectrometric technique.

6) There is few details on the instruments used for the analyses, such as IR, ESI-MS and TGA (e.g. name and state of the manufacturer..). Also, there should be more details on the analytical methodology used for the analyses .

7) On page 18, EPR is written incorrectly ("ERP").

8) There are no doi in many articles in the references section.

I have few questions for the Authors:

Q1) It would be interesting to be able to check the IR spectra and the ESI+-MS mass spectra of the compounds. Can you add them in the ESI file?

D2) The theranostic compound Pluvicto(R) (Lutetium 177Lu vipivotide tetraxetan → here an article about it: https://www.targetedonc.com/view/pluvicto-approval-launches-theranostics-in-prostate-cancer) was recently approved by the FDA for medical use: Could you add this reference in the Introduction part, particularly when you specifying that there are currently no clinically-approved theranostic agents?

In conclusion, I recommend the publication of the paper in Phamaceutics after minor revisions. 

Author Response

Dear Reviewer!

Thank you very much for your time to evaluate our manuscript and for your valuable comments, which we considered all as follows:

Review comments

There are just small changes to the English and style of the manuscript to be made by the Authors, e.g .:

1) "in vitro" and "in vivo" must be written in italics;

2) Drugs names such as Magnevist... should be written with the copyright symbol "(R)" at the end of the word (in superscript). The same goes for ultrapure water Milli-Q .

3) Check the "numbers" in the chemical formulas: many are not written in subscript. Same goes for the charge state of the ions (in superscript) (see page 2 for example).

4) The parameters "T1, T2, T2 *, r1" should be written in italics (the letter) and in subscript (the number);

5) The ESI-MS method is not of the spectroscopic type but it is a spectrometric technique.

6) There is few details on the instruments used for the analyses, such as IR, ESI-MS and TGA (e.g. name and state of the manufacturer..). Also, there should be more details on the analytical methodology used for the analyses .

7) On page 18, EPR is written incorrectly ("ERP").

8) There are no doi in many articles in the references section.

Response to points 1-8: We revised thoroughly the whole manuscript and made all changes and suggestions as to the reviewer’s comments. The revised manuscript was checked and edited by a native English speaker, Mrs. Mary McAllister too.

I have few questions for the Authors:

Q1) It would be interesting to be able to check the IR spectra and the ESI+-MS mass spectra of the compounds. Can you add them in the ESI file?

Response to Q1: The recorded IR and ESI-MS spectra have been included in the Supplementary File. The methods’ descriptions and the results have been extensively reported and discussed (see revised Manuscript and revised Supplementary File).

D2) The theranostic compound Pluvicto(R) (Lutetium 177Lu vipivotide tetraxetan → here an article about it: https://www.targetedonc.com/view/pluvicto-approval-launches-theranostics-in-prostate-cancer) was recently approved by the FDA for medical use: Could you add this reference in the Introduction part, particularly when you specifying that there are currently no clinically-approved theranostic agents?

Response to D2: We revised thoroughly the introductory part of our manuscript and cited the only one recently approved theranostic agent for the treatment of progressive prostate cancer Pluvicto(R)  (see revised Manuscript).

Author Response

Dear Reviewer!

Thank you very much for your time to evaluate our manuscript and for your valuable comments, which we considered all as follows:

Review comments

  1. the introductory part of the article should be written in a way that better supports the present results.

Response to point 1: The introduction has been thoroughly revised, including the only one recently approved theranostic agent for the treatment of progressive prostate cancer and pointing out the potential advantages of the presented work (see revised Manuscript).

   2. the figures must also be summarised to reduce the number of diagrams and facilitate comparison.

Response to point 2: The following figures in the manuscript have been revised as follows: Figures 1, 2, 3, 5 and 6 have been overworked to present the structure and physicochemical properties of the new Sal-based complexes in the best and comprehensive way; Figure 4 - all R1 plots of the free Sal-complexes and the two control used as well as the R1 plots of the BGs formulations have been combined respectively for a better comparison. One point from the Sal-Gd(III) graph was omitted on purpose as it lied out of the range when related to the other graphs (see revised Manuscript).

3.In the introduction of the article, it is mentioned that Gd has a toxic effect, so we choose a Mn-based complex for imaging.

  1. You need to make an FTIR diagram.

Response to point 3.1: We have already carried out IR and ESI-MS measurements to elucidate the structure of the new Sal-based complexes. The method description and the results have been extensively reported and discussed (see revised Manuscript). IR and ESI-MS spectra have been now included in the Supplementary File (see revised Supplementary File).

     2. Relaxation time is enough to justify its effect as a good contrast agent. Is there any in vivo imaging to confirm the result. Even Mn-based conjugates showed shorter relaxation time.

Response to point 3.2: So far we do not have any in vivo results with the newly prepared Sal-based compounds as this is the first of a series of papers on the development of new Sal-based theranostics. However, based on our extensive and promising preparatory work on macromolecular polysaccharide-based conjugates with Mn(II) (manuscript in preparation) as well as on the solid reported work on novel Mn(II)-based contrast agents for MRI as alternatives to Gd-based ones (Zhou, I. Y., Ramsay, I. A., Ay, I., Pantazopoulos, P., Rotile, N. J., Wong, A., Caravan, P., & Gale, E. M. (2021). Positron Emission Tomography-Magnetic Resonance Imaging Pharmacokinetics, In Vivo Biodistribution, and Whole-Body Elimination of Mn-PyC3A. Investigative radiology56(4), 261–270. https://doi.org/10.1097/RLI.0000000000000736; Garda, Z., Forgács, A., Do, Q. N., Kálmán, F. K., Timári, S., Baranyai, Z., Tei, L., Tóth, I., Kovács, Z., & Tircsó, G. (2016). Physico-chemical properties of MnII complexes formed with cis- and trans-DO2A: thermodynamic, electrochemical and kinetic studies. Journal of inorganic biochemistry163, 206–213. https://doi.org/10.1016/j.jinorgbio.2016.07.018; Sguizzato, M., Martini, P., Marvelli, L., Pula, W., Drechsler, M., Capozza, M., Terreno, E., Del Bianco, L., Spizzo, F., Cortesi, R., & Boschi, A. (2022). Synthetic and Nanotechnological Approaches for a Diagnostic Use of Manganese. Molecules (Basel, Switzerland)27(10), 3124. https://doi.org/10.3390/molecules27103124; ), we believe that our planned in vivo studies will show a good signal enhancement in MRI as the in vitro results did.

In addition, Mn-based MRI contrast compounds, notably the recently developed Mn-PyC3A, show a similar performance as Gd and have an excellent safety profile (Zhou, I. Y., Ramsay, I. A., Ay, I., Pantazopoulos, P., Rotile, N. J., Wong, A., Caravan, P., & Gale, E. M. (2021). Positron Emission Tomography-Magnetic Resonance Imaging Pharmacokinetics, In Vivo Biodistribution, and Whole-Body Elimination of Mn-PyC3A. Investigative radiology56(4), 261–270. https://doi.org/10.1097/RLI.0000000000000736).

As our Mn-based compounds are macromolecular ones with the paramagnetic center surrounded by the Sal chelator as in a cage, we believe in additional MRI enhancing effect due to lower molecular tumbling of the contrast agent.

       3. The aim of these compounds is to be used for diagnostic purposes and it shows the toxic effect on cells. Even though the authors used cancer cells, the goal here is to show non-toxicity. Here the SalMn conjugate has a higher toxicity. The authors need to do different tests to check the significant effect on toxicity.

Response to point 3.3: With regard to toxicity of the novel Sal-based compounds, we expect a negligible or even no toxicity for the following reasons: 1) The successful encapsulation of both Sal-Gd(III) and Sal-Mn(II) into Bacterial Ghosts Cells (BGs), shows preserved anticancer efficacy and even a stronger signal enhancement in MRI at lower concentrations of the entrapped complexes; 2) Vast scientific research reports on the selective mode of action of Sal alone, where only the cancer cells have been eliminated and no healthy ones (Antoszczak M. (2019). A comprehensive review of salinomycin derivatives as potent anticancer and anti-CSCs agents. European journal of medicinal chemistry166, 48–64. https://doi.org/10.1016/j.ejmech.2019.01.034; Antoszczak, M., & HuczyÅ„ski, A. (2019). Salinomycin and its derivatives - A new class of multiple-targeted "magic bullets". European journal of medicinal chemistry176, 208–227. https://doi.org/10.1016/j.ejmech.2019.05.031; Dewangan, J., Srivastava, S., & Rath, S. K. (2017). Salinomycin: A new paradigm in cancer therapy. Tumour biology : the journal of the International Society for Oncodevelopmental Biology and Medicine39(3), 1010428317695035. https://doi.org/10.1177/1010428317695035);

Besides the predominant anticancer activity of Sal, it has been also verified that it does not emerge severe adverse effects on human normal tissues like other conventional chemotherapeutical drugs. For instance, Sal induces T-cells apoptosis in T-lymphocytic leukemia patients, but not in healthy people (Fuchs, D., Heinold, A., Opelz, G., Daniel, V., & Naujokat, C. (2009). Salinomycin induces apoptosis and overcomes apoptosis resistance in human cancer cells. Biochemical and biophysical research communications390(3), 743–749. https://doi.org/10.1016/j.bbrc.2009.10.042).

We have shown recently no toxicity and even total recovery in murine animal models when applying BGs loaded either with 5-Fluorouracil (5FU), 5-Fluorouridine, Oxaliplatin (Oxa) or with Doxorubicin (Dox) (all clinically applied anticancer drugs with severe side effects) (manuscripts in preparation). All newly prepared BG-formulations exerted superior anticancer activity that led to prolonged overall survival (when loaded with 5FU or Dox) of the tested animals and tumor-free mice (with loaded Oxa).

The present work is only the first of a series of papers reporting on the preliminary evaluation of the potential of the new compounds as theranostic agents. Thus, we focused here on the preparation and on the detailed characterization of the imaging probes. As next and as a part of an ongoing project, we have planned in vivo testing starting from April 2023. For this we already have an ethic approval with the license number 66.009/0284-WF/V/3b/2017. Moreover, we are preparing currently an application for additional funding to elucidate also the tolerable LD50 of all new Sal-based compounds.

Round 2

Reviewer 2 Report

The authors reply to the comments and the modifications are convincing.